# The Multiquadric Kernel for Moment-Matching Distributional Reinforcement Learning

**Ludvig Killingberg**                                                                    *ludvig.killingberg@ntnu.no*
*Department of Computer Science*
*Norwegian University of Science and Technology*

**Helge Langseth**                                                                      *helge.langseth@ntnu.no*
*Department of Computer Science*
*Norwegian University of Science and Technology CIFAR Fellow*

**Reviewed on OpenReview:** *https://openreview.net/forum?id=z49eaB8kiH*

## Abstract

Distributional reinforcement learning has gained significant attention in recent years due to its ability to handle uncertainty and variability in the returns an agent will receive for each action it takes. A key challenge in distributional reinforcement learning is finding a measure of the difference between two distributions that is well-suited for use with the distributional Bellman operator, a function that takes in a value distribution and produces a modified distribution based on the agent's current state and action. In this paper, we address this challenge by introducing the multiquadric kernel to moment-matching distributional reinforcement learning. We show that this kernel is both theoretically sound and empirically effective. Our contribution is mainly of a theoretical nature, presenting the first formally sound kernel for moment-matching distributional reinforcement learning with good practical performance. We also provide insights into why the RBF kernel has been shown to provide good practical results despite its theoretical problems. Finally, we evaluate the performance of our kernel on a number of standard benchmarks, obtaining results comparable to the state-of-the-art.

## 1  Introduction

Reinforcement learning is a type of machine learning that involves training agents to take actions in an environment in order to maximize a reward signal. In traditional reinforcement learning, the agent learns a value function that estimates the expected future cumulative discounted reward (value) for each state-action pair. The value function tells the agent how "good" a particular state-action pair is, and the agent uses this information to make decisions about which actions to take in order to maximize its value. However, this approach can be problematic in some cases, such as when the environment is highly stochastic or when the reward signal is noisy.

Distributional reinforcement learning is a variation of reinforcement learning that addresses these challenges by learning a distribution of future cumulative discounted rewards (value distribution), rather than a single expected value, for each state-action pair. This distributional approach allows the agent to consider the uncertainty or variability in the rewards it may receive, and can lead to more robust and accurate decision-making.

Algorithms for distributional reinforcement learning can be separated along two axis: the representation of the value distribution for a given state-action pair and the measurement of the difference between distributions. Some algorithms utilize a discrete set of samples to represent the distribution (Bellemare et al., 2017; Dabney et al., 2017; Nguyen-Tang et al., 2021), while others employ implicit distributions parameterized by a neural network (Dabney et al., 2018; Yang et al., 2019). Additionally, the agent must have the ability to measure the difference between the predicted value distribution for a given state-action pair and the actual value distribution received, in order to update its predictions. This difference measure is crucial for the agent to learn and adapt. The Kullbach-Leibler divergence is widespread in the probabilistic machine learning field but possesses some undesirable properties for use in distributional reinforcement learning. Instead, measures such as the Wasserstein metric, the Cramér distance, and the maximum mean discrepancy

have shown better results (Bellemare et al., 2017; Dabney et al., 2017; Nguyen-Tang et al., 2021; Dabney et al., 2018; Yang et al., 2019).

The distributional Bellman operator is a key component of distributional reinforcement learning algorithms. It is used to update the value distribution for a given state-action pair based on the observed rewards and transitions in the environment. The distributional Bellman operator is a contraction under certain conditions, which means that it guarantees that the distance between the updated and original value distribution will always be non-increasing, converging to the true value distribution in the limit (with convergence properties similar to those of the usual Bellman operator used, e.g., in Q-learning).

Our main contribution in this paper is a theoretical analysis and extension of the work by Nguyen-Tang et al. (2021) on on moment matching for distributional RL. Additionally, our results may also be of theoretical interest for understanding the properties of the distributional Bellman operator.

## 2 Background

This section introduces the necessary background in both classical reinforcement learning, and distributional reinforcement learning, and describe some current state-of-the-art distributional reinforcement learning methods.

### 2.1 Reinforcement Learning

We consider the standard reinforcement learning (RL) setting, where interaction between an agent and its environment is modeled as a Markov Decision Process $(\mathcal{S}, \mathcal{A}, R, P, \gamma)$ (Puterman, 1994), where $\mathcal{S} \subseteq \mathbb{R}^d$ and $\mathcal{A}$ denote the state and action space respectively. The stochastic reward function $R : \mathcal{S} \times \mathcal{A} \to \mathbb{R}$ is a function that maps the state and action of an agent to a reward value with some element of randomness or uncertainty. $P(\cdot|\boldsymbol{x}, a)$ denotes the transition probabilities starting in state $\boldsymbol{x} \in \mathcal{S}$ and doing $a \in \mathcal{A}$, and $\gamma \in [0, 1)$ is the discount factor. A policy $\pi(\cdot|\boldsymbol{x})$ maps states to distributions over actions.

The Bellman equation is often used as the foundation for algorithms that solve MDPs, such as Q-learning. In general, the equation describes a necessary condition for an optimal solution in dynamic programming optimization. In the context of reinforcement learning, it is often used to describe the optimal state-action value function $Q^*$. According to the Bellman equation, described by (1), the accumulated discounted reward, or value, of a state-action pair is equal to the sum of the immediate reward from the current action and the discounted expected reward from future actions:

$$Q^*(\boldsymbol{x}, a) = \mathbb{E}\left[R(\boldsymbol{x}, a)\right] + \gamma \mathbb{E}_{\boldsymbol{x}' \sim P}\left[\max_{a' \in \mathcal{A}} Q^*(\boldsymbol{x}', a')\right]. \tag{1}$$

A common approach to solving the Bellman equation uses the Bellman operator. The Bellman operator, $\mathcal{T}^\pi$, maps state-action values to their next state-action values. The Bellman operator for a policy $\pi$ is defined as follows:

$$\mathcal{T}^\pi Q(\boldsymbol{x}, a) := \mathbb{E}\left[R(\boldsymbol{x}, a)\right] + \gamma \mathbb{E}_{\boldsymbol{x}' \sim P, a' \sim \pi}\left[Q(\boldsymbol{x}', a^\pi)\right]. \tag{2}$$

Alternatively, we can define the Bellman operator for an optimal policy, also known as the Bellman optimality operator as

$$\mathcal{T}^* Q(\boldsymbol{x}, a) := \mathbb{E}\left[R(\boldsymbol{x}, a)\right] + \gamma \mathbb{E}_{\boldsymbol{x}' \sim P}\left[\max_{a' \in \mathcal{A}} Q(\boldsymbol{x}', a')\right]. \tag{3}$$

The significance of the Bellman operators is that they are contraction mappings wrt the supremum norm. In other terms, for some $k \in (0, 1)$, we have

$$\left\|\mathcal{T}^\pi Q_1 - \mathcal{T}^\pi Q_2\right\|_\infty \leq k \left\|Q_1 - Q_2\right\|_\infty. \tag{4}$$

According to the Banach fixed-point theorem Banach (1922), that means that we have the following properties:

$$\lim_{k \to \infty} (\mathcal{T}^\pi)^k Q = Q^\pi, \tag{5}$$

$$\lim_{k \to \infty} (\mathcal{T}^*)^k Q = Q^*. \tag{6}$$

Deep Q-Network (DQN) (Mnih et al., 2013) approximates $Q$ using a neural network, which we will denote $Q_\theta$. Mnih et al. (2013) proceed to approximate $Q^*$ using gradient descent to minimize the distance

$$L(\theta) = d\left(\mathcal{T}^\pi Q_\theta(\boldsymbol{x}_t, a_t), Q_\theta(\boldsymbol{x}_t, a_t)\right). \tag{7}$$

More specifically, they let $d$ be the $\ell_2$ distance, in which case this is also called the squared temporal difference.

## 2.2 Distributional Reinforcement Learning

Distributional reinforcement learning focuses on learning the distribution of cumulative discounted rewards, rather than just the expected value. In distributional reinforcement learning, the algorithm typically uses a function to approximate the distribution over returns for each state-action pair. This function is updated over time based on the rewards received and the predicted distribution of rewards for each state-action pair. By doing this, the algorithm can learn to anticipate the range of possible cumulative discounted rewards and make decisions that maximize the expected value.

For an agent following policy $\pi$, $Z^\pi$, or *value distribution* (Bellemare et al., 2017), is the sum of discounted rewards along the agent's trajectory:

$$Z^\pi(\boldsymbol{x}, a) := \sum_{t=0}^{\infty} \gamma^t R(\boldsymbol{x}_t, a_t), \tag{8}$$

starting in state $\boldsymbol{x}_0 = \boldsymbol{x}$, with $a_0 = a$, $a_t \sim \pi(\cdot|\boldsymbol{x}_t)$, and $\boldsymbol{x}_t \sim P(\cdot|\boldsymbol{x}_{t-1}, a_{t-1})$. The objective of RL can be summarized as finding the optimal policy, that is the sequence of actions that an agent should take, in order to maximize the reward signal $Q^\pi(\boldsymbol{x}, a) := \mathbb{E}_{P,R,\pi}\left[Z^\pi(\boldsymbol{x}, a)\right]$ of a given state-action pair. In other words, we look for the strategy that the agent should follow to achieve the best possible outcome in expectation.

The distributional Bellman operator is a mathematical operator used in distributional reinforcement learning to estimate the probability distribution of future rewards for each state-action pair in an environment. This operator is based on the Bellman equation. The distributional Bellman operator extends the Bellman equation to estimate the probability distribution of future rewards, rather than just the expected value. This allows the operator to capture the uncertainty and variability of the rewards in the environment, and to make predictions about the range of possible outcomes. Given a random variable $Z^\pi$ denoting the value distribution, the operator is defined as

$$\mathcal{T}^\pi Z^\pi(\boldsymbol{x}, a) \stackrel{D}{:=} R(\boldsymbol{x}, a) + \gamma Z^\pi(\boldsymbol{x}', a'), \tag{9}$$

where $A \stackrel{D}{=} B$ denotes equality in distribution, i.e. random variable $A$ and $B$ have the same distribution, and $a' \sim \pi, \boldsymbol{x}' \sim P(\boldsymbol{x}'|\boldsymbol{x}, a')$.

The distributional version of Equation 7 is

$$L(\theta) = d\left(\mathcal{T}^\pi Z_\theta(\boldsymbol{x}_t, a_t), Z_\theta(\boldsymbol{x}_t, a_t)\right), \tag{10}$$

but in this case, because we are operating on distributions and not values, we can no longer use the $\ell_2$ metric. There are many distance measures for distributions to choose between, but not all of them are suitable for this use case. It is important for distributional Q-learning that the distributional Bellman operator is a contraction in $d$. What this means is that for some $k \in (0, 1)$, we have that

$$d\left(\mathcal{T}^{\pi}Z_1, \mathcal{T}^{\pi}Z_2\right) \leq kd\left(Z_1, Z_2\right). \tag{11}$$

We can then use $d$, or some estimator of $d$, as a loss function, and our algorithm will converge.

Bellemare et al. (2017) show that the distributional Bellman operator for a fixed policy ($\mathcal{T}^{\pi}$) is a contraction in the $p$-Wasserstein metric and Cramér distance, but not in KL-divergence, total variation, or Kolmogorov distance. Nguyen-Tang et al. (2021) further show that $\mathcal{T}^{\pi}$ is a contraction in maximum mean discrepancy for carefully selected kernels.

### 2.2.1 Learning Categorical Distributions

One popular approach to distributional reinforcement learning is to approximate the distribution of values, $Z$, by a discretized or categorical variable. C51 was introduced by Bellemare et al. (2017) as an extension of DQN. Like DQN, C51 uses a neural network to approximate the action-value function, but instead of outputting a single value for each action, it outputs a distribution over the possible returns. This distribution is represented using a set of discrete "atoms", which allows the agent to represent a wide range of possible returns. 51 discrete atoms per action were used in their Atari 2600 experiments, hence the name C51.

In C51, the distance between two distributions is calculated using the Kullback-Leibler (KL) divergence. One issue with using the KL divergence as a measure of the difference between two distributions in distributional reinforcement learning is that the distributional Bellman operator is not a contraction in KL divergence (Bellemare et al., 2017). Because the Bellman operator is not a contraction in KL-divergence, there is no fixed point, hence it cannot be used for TD learning.

For this reason, KL divergence is generally not considered a good measure of the difference between two distributions in distributional reinforcement learning. Instead, other methods typically use measures such as the Wasserstein distance or the Cramér distance, which do satisfy the contraction property Bellemare et al. (2017).

### 2.2.2 Quantile Regression

Quantile Regression DQN (QR-DQN) is a variant of DQN introduced by Dabney et al. (2017). To learn the value distribution, QR-DQN uses quantile regression Q-learning, which is based on quantile regression, a statistical technique for estimating the conditional quantile function of a random variable.

One advantage of QR-DQN is that it uses the Wasserstein distance, also known as the Earth Mover's distance, as a measure of the difference between two distributions. The Wasserstein distance is a well-known metric in the field of probability theory and statistics, and it has several desirable properties that make it a good choice for distributional reinforcement learning:

- The Wasserstein distance is a true metric, meaning that it satisfies the triangle inequality and is symmetric. This makes it a more consistent and reliable measure of the difference between two distributions than the Kullback-Leibler (KL) divergence, which does not satisfy these properties.

- The Wasserstein distance is a smooth, continuous function, which makes it well-suited for use with gradient-based optimization methods. This is important in reinforcement learning, where algorithms often rely on gradient-based optimization to learn the value function.

- The distributional Bellman operator is a contraction in Wasserstein distance.

### 2.2.3 Implicit Quantile Regression

Implicit quantile regression DQN (IQR-DQN) (Dabney et al., 2018) is a variant of the QR-DQN (Dabney et al., 2017) algorithm. Unlike QR-DQN, which uses an explicit quantile regression model to estimate the distribution of returns, IQR-DQN uses an implicit quantile model, which is a neural network that directly maps states and actions to quantiles of the return distribution.

Work on IQR-DQN (Dabney et al., 2018; Yang et al., 2019) has focused on developing efficient and effective algorithms for learning the implicit quantile model, as well as on demonstrating the performance of IQR-DQN on a variety

of reinforcement learning tasks. Results have shown that IQR-DQN can learn more accurate value distributions than other distributional reinforcement learning methods and that it can achieve better performance.

### 2.2.4 Moment-matching

Nguyen-Tang et al. (2021) proposed to minimize the maximum mean discrepancy (MMD), rather than the Wasserstein distance. The squared maximum mean discrepancy between two distributions $p$ and $q$ relative to a kernel $k$ is defined as

$$\text{MMD}^2(p, q; k) = \mathbb{E}\left[k(X, X')\right] + \mathbb{E}\left[k(Y, Y')\right] - 2\mathbb{E}\left[k(X, Y)\right], \tag{12}$$

where $X, X'$ are independent random variables with distribution $p$ and $Y, Y'$ are independent random variables with distribution $q$. Given two sets of samples $\{\boldsymbol{x}_i\}_{i=1}^m \sim p$ and $\{\boldsymbol{y}_i\}_{i=1}^m \sim q$, Gretton et al. (2012) introduced the following unbiased estimator of MMD

$$\widehat{\text{MMD}}_u^2(\{\boldsymbol{x}_i\}, \{\boldsymbol{y}_i\}; k) := \frac{1}{m(m-1)} \sum_{i \neq j} k(\boldsymbol{x}_i, \boldsymbol{x}_j) + k(\boldsymbol{y}_i, \boldsymbol{y}_j) - 2k(\boldsymbol{x}_i, \boldsymbol{y}_j).$$

Nguyen-Tang et al. (2021) use a biased estimator $\widehat{\text{MMD}}_b$ of MMD, but justify the choice by noting that the estimator in practice shows lower variance than its unbiased counterpart. This biased estimator, also introduced by Gretton et al. (2012), is defined as

$$\widehat{\text{MMD}}_b^2(\{\boldsymbol{x}_i\}, \{\boldsymbol{y}_i\}; k) := \frac{1}{m^2} \sum_{i,j} k(\boldsymbol{x}_i, \boldsymbol{x}_j) + k(\boldsymbol{y}_i, \boldsymbol{y}_j) - 2k(\boldsymbol{x}_i, \boldsymbol{y}_j).$$

**Kernels** One set of interesting kernels are those on the form $k(\boldsymbol{x}, \boldsymbol{y}) = -\|\boldsymbol{x} - \boldsymbol{y}\|_2^\beta$. Nguyen-Tang et al. (2021) call this class of kernels unrectified kernels. MMD with the unrectified kernel and $\beta = 2$ is zero as long as the two distributions are equal in expectation (Székely & Rizzo, 2013). This means that we will not see the convergence in distribution, but rather in expectation, essentially reducing it to classical Q-learning. Nguyen-Tang et al. (2021) look at the practical performance of this kernel, and finds it performs unfavorably compared to some other kernels.

Székely & Rizzo (2013) showed that with kernels of the form $k(\boldsymbol{x}, \boldsymbol{y}) = -\|\boldsymbol{x} - \boldsymbol{y}\|_2^\beta$ for $\beta \in (0, 2)$, then $\text{MMD}(\mu, \nu; k)$ is zero if and only if $\mu$ and $\nu$ are equal in distribution. Nguyen-Tang et al. (2021) further show that the distributional Bellman operator is a contraction in these kernels. Practical performance was evaluated for $\beta = 1$, but the results were not promising.

The RBF kernel is a common choice in Gaussian processes and other applications. It is defined as

$$k_h(\boldsymbol{x}, \boldsymbol{y}) = \exp\left(-h^2 \|\boldsymbol{x} - \boldsymbol{y}\|_2^2\right), \tag{13}$$

where $h$ is a free parameter. Nguyen-Tang et al. (2021) showed by counterexample that the distributional Bellman operator is not a contraction under MMD with the RBF kernel, but also found that the RBF kernel performs favorably compared to the unrectified kernels when using a mixture of RBF kernels with different length-scale values, defined as

$$k(\boldsymbol{x}, \boldsymbol{y}) = \sum_{l=1}^{10} \exp\left(-l^{-1} \|\boldsymbol{x} - \boldsymbol{y}\|_2^2\right). \tag{14}$$

In this paper, we will investigate the performance of the multiquadric kernel (MQ) (Hardy, 1971) for moment-matching distributional reinforcement learning. Given a free parameter $h$, the multiquadric kernel is defined as

$$k_h(\boldsymbol{x}, \boldsymbol{y}) = -\sqrt{1 + h^2 \|\boldsymbol{x} - \boldsymbol{y}\|_2^2}. \tag{15}$$

| Radial Kernel | $\psi(t)$ | 1 | 2 | 3 | 4 |
|---|---|---|---|---|---|
| Multiquadric | $-\sqrt{1+h^2t^2}$ | ✓ | ✓ | ✓ | ✓ |
| RBF | $\exp(-h^2t^2)$ | ✓ | ✗ | ✓ | ✓ |
| Unrectified | $-\lvert t\rvert$ | ✓ | ✓ | ✗ | ✓ |
| Unrectified | $-t^2$ | ✗ | ✓ | ✓ | ✗ |

Table 1: Property 1 - 4 for various radial kernels.

Note that the term kernel is often used as a shorthand for positive-definite kernels. We will not make this restriction and instead, refer to a kernel as any symmetric real-valued function of two variables. The MQ, unlike the RBF, is not positive definite but is in a class called *conditionally* positive definite. In Section 3 we look at the properties of this kernel.

**Pseudo-samples**  MMDQN (Nguyen-Tang et al., 2021) use the QR-DQN network structure to approximate $Z$. They define the $N$ outputs per action to be pseudo-samples from $Z_\theta$, and then minimize $\mathrm{MMD}_b^2$ between the pseudo-samples from $Z_\theta$ and $\mathcal{T}Z_\theta$. This means that true samples from the value distribution are never drawn, and that the approximation is completely deterministic. There are no underlying restrictions with the algorithm that would suggest that a sampling-based approach would not also work.

**Open question**  Nguyen-Tang et al. (2021) pose a question on what the sufficient conditions on the kernels are for the Bellman operator to be a contraction in MMD. They further question whether scale sensitivity is a necessary condition. In Section 3 we show by counterexample that scale sensitivity is not a necessary condition for contraction in MMD.

## 3 Theoretical Work

In this paper, we propose to use the multiquadric kernel for moment-matching distributional reinforcement learning. Figure 1 provides an illustrative comparison between the multiquadric kernel and the kernels discussed in section 2.2.4. Our choice of the multiquadric kernel, $k_h$, is motivated by several desirable properties it possesses:

1. $\mathrm{MMD}(\cdot,\cdot;k_h)$ is a metric on probability distributions,

2. The Bellman operator is a contraction in MMD with $k_h$,

3. $\mathrm{MMD}_b(\cdot,\cdot;k_h)$ is smooth,

4. $\sup_{\boldsymbol{x}\in\mathbb{R}^n}\lVert\frac{\partial}{\partial\boldsymbol{x}}\mathrm{MMD}_b^2(\boldsymbol{x},\boldsymbol{y};k_h)\rVert_2\leq C_h<\infty$ for some constant $C_h$ dependent on $h$.

Table 1 shows property 1 - 4 for a range of radial kernels, i.e. kernels on the form $k(\boldsymbol{x},\boldsymbol{y})=\psi(\lVert\boldsymbol{x}-\boldsymbol{y}\rVert_2)$. We observe that only the multiquadric kernel possesses all the aforementioned properties. As far as we know, the MQ kernel is the only kernel that satisfies all properties 1 - 4 and is the first non-scale-sensitive kernel with the contraction property. Proofs for property 1 and 2 will follow. Proofs for property 3 and 4 are available in the appendix as Lemma A.1 and Lemma A.2 respectively.

First, we show that under the multiquadric kernel, maximum mean discrepancy (MMD) is a metric in the family of valid distributions. In other words, it satisfies the triangle inequality, is symmetric, non-negative, and is zero if and only if the distributions are equal. This is important in the context of distributional reinforcement learning, as it ensures that the distance measure used to compare the predicted and true distributions of returns is consistent and reliable.

Second, we demonstrate that the distributional Bellman operator is a contraction in MMD with the multiquadric kernel. This property is important in reinforcement learning, as it ensures that the agent's estimate of the distribution of returns will necessarily get closer to the true distribution as it takes more actions and gathers more experiences. This can help the agent to more accurately learn the distribution of returns and make good decisions based on that distribution.

Furthermore, the multiquadric kernel has smoothness properties, resulting in a smooth maximum mean discrepancy when using the multiquadric kernel. In many practical scenarios, it is commonly observed that a smoothness property

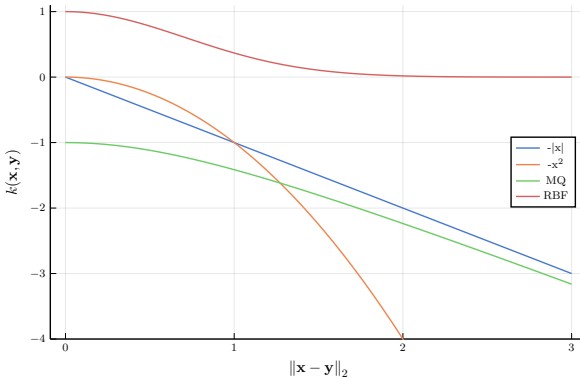

Figure 1: The multiquadric (MQ) kernel compared to the kernels discussed in section 2.2.4 as a function of the Euclidean distance between two points. $h = 1$ was chosen for both MQ and RBF.

in the loss function tends to lead to gradients that exhibit a greater degree of stability and better performance (Gokcesu & Gokcesu, 2021; Patterson et al., 2023). In the context of reinforcement learning, this can ultimately lead to faster convergence and better performance of the model. This is one of the motivations behind the use of the $\ell_2$ metric and the Huber loss over $\ell_1$ metric in deep learning.

Finally, the magnitude of the gradient of the kernel is upper bounded by a constant. If the magnitude of the gradient is not upper-bounded, the learning algorithm may oscillate or diverge, rather than converge to a stable and accurate solution. By upper bounding the magnitude of the gradient, it is possible to ensure that the learning algorithm converges to a stable and accurate solution in a reasonable amount of time. This is one of the reasons the Huber loss has become so popular in reinforcement learning, and why we often see gradient clipping being performed when the $\ell_2$ metric is used as a loss function.

Our proof that the distributional Bellman operator is a contraction in MMD with the multiquadric kernel requires a few definitions and existing lemmas. We begin by defining *completely monotonic functions* and *conditionally (strictly) positive definite kernels*.

**Definition 3.1.** $f(x)$ is a completely monotonic function (Schilling et al., 2012, Definition 1.3) if it is infinitely differentiable and

$$(-1)^n f^{(n)}(x) \geq 0 \text{ for } x > 0 \text{ and } n = 0, 1, 2, \dots$$

**Definition 3.2.** A real-valued kernel $k$ is called *conditionally* positive definite if it is symmetric, i.e. $k(\boldsymbol{x}, \boldsymbol{y}) = k(\boldsymbol{y}, \boldsymbol{x})$, and satisfies

$$\sum_{i=1}^{m} \sum_{j=1}^{m} c_i c_j k(\boldsymbol{x}_i, \boldsymbol{x}_j) \geq 0 \tag{16}$$

for any $\boldsymbol{c} \in \mathbb{R}^m$ with

$$\sum_{i=1}^{m} c_i = 0.$$

It is called conditionally *strictly* positive definite if the greater-than or equal sign in Equation 16 can be replaced by a greater-than sign.

The following three lemmas provide a basis for our main results in Theorem 3.1 and Theorem 3.2.

**Lemma 3.1** (Székely & Rizzo, 2013, Proposition 3). $\text{MMD}(\cdot, \cdot; k)$ is a metric on $\mathcal{P}(\mathcal{X})$ if and only if $k$ is a conditionally strictly positive definite kernel.

**Lemma 3.2** (Micchelli, 1986, Theorem 2.1). Let $g(t) := f(\sqrt{t})$. Then $f$ is conditionally positive definite if and only if $-g'(t)$ is completely monotonic.

**Lemma 3.3** (Wendland, 2004, Corollary 8.20). $f$ is conditionally *strictly* positive definite if and only if it is conditionally positive definite and not a polynomial of order 2 or less.

Theorem 3.1 formally states property 1 of the multiquadric kernel.

**Theorem 3.1.** $\mathrm{MMD}(\cdot, \cdot; k_h)$ is a metric on $\mathcal{P}(\mathcal{X})$ for the multiquadric kernel $k_h(\boldsymbol{x}, \boldsymbol{y}) = -\sqrt{1 + h^2 \|\boldsymbol{x} - \boldsymbol{y}\|_2^2}$

*Proof.* By utilizing Lemma 3.1, we only need to show that the multiquadric kernel is conditionally strictly positive definite.

We rewrite $k_h(\boldsymbol{x}, \boldsymbol{y})$ as $f(h^2 \|\boldsymbol{x} - \boldsymbol{y}\|_2)$. We can directly show that for $g(t) = f(\sqrt{t}) = -\sqrt{1+t}$ with $t > 0$, then $-g'(t) = \frac{1}{2\sqrt{1+t}}$ is completely monotonic. By induction, we see that its $n$-th derivative

$$\frac{d^n}{dt^n} \frac{1}{2\sqrt{1+t}} = \frac{\sqrt{\pi}(x+1)^{-n-1/2}}{2\Gamma\left(\frac{1}{2} - n\right)}$$

has an oscillating sign on $(0, \infty)$, due to the sign of the gamma function for strictly negative values. By Lemma 3.2, we conclude that $f(t)$ is conditionally positive definite. By Lemma 3.3, $f(t)$ is conditionally strictly positive definite. $\square$

Lemma 3.4 is the basis for our proof that the distributional Bellman operator is a contraction in MMD with the multiquadric kernel (Theorem 3.2).

**Lemma 3.4.** $f(t; a, b) := \sqrt{a + t^2} - \sqrt{b + t^2}$ is positive definite for $a > b$.

To establish the positive definiteness of $f(t)$, we begin by taking the $n$-th derivative of $f(\sqrt{t})$ and demonstrating that it is completely monotonic. Utilizing Lemma 3.4, we can then conclude that $f(t)$ is positive definite. A comprehensive proof can be found in the appendix.

**Theorem 3.2.** The distributional Bellman operator is a contraction in MMD with the multiquadric kernel.

The detailed proof of Theorem 3.2 is given in the appendix. The main step is to show that it is sufficient to prove that the kernel,

$$k(\boldsymbol{x}, \boldsymbol{y}) = \sqrt{\frac{1}{\gamma^2} + h^2 \|\boldsymbol{x} - \boldsymbol{y}\|_2^2} - \sqrt{1 + h^2 \|\boldsymbol{x} - \boldsymbol{y}\|_2^2},$$

is conditionally positive definite. As a result of Lemma 3.4 the kernel is positive definite; hence, it is also conditionally positive definite.

# 4 Limitations of the Radial Basis Function for MMD

In this section, our aim is to investigate the sensitivity of MMDQN (Nguyen-Tang et al., 2021) performance on Atari problems to RBF kernel parameters. While MMDQN has achieved remarkable performance across all 57 games, questions remain regarding how sensitive this performance is to changes of the kernel. The use of a single mixture kernel for all 57 games, may give the impression that the method is relatively insensitive to the kernel, however, we will show that this is not the case. We will also identify settings where the optimal RBF parameters are more difficult to find. It is worth noting that the authors do not claim that these parameters will generalize to different environments; quite the contrary, they state that the kernel's role in performance is crucial. They also state that identifying a kernel that performs well in all games is challenging. In this context, we explore the impact of the RBF bandwidth parameter on MMDQN performance in simple toy settings.

It is interesting to see whether there exist settings in which the RBF kernel does possess the contraction property. Using a similar proof to that of Theorem 3.2, we can show that for two distributions $\nu, \mu$ with bounded support, the distributional Bellman operator is a contraction in MMD with the RBF kernel under certain bounds for $h$. Corollary 3.1 defines the bounds. The proof is given in the appendix.

**Corollary 4.1.** Let $k(\boldsymbol{x}, \boldsymbol{y}) = \exp(-h^2 \|\boldsymbol{x} - \boldsymbol{y}\|_2^2)$. Then the distributional Bellman operator is a contraction in $\mathrm{MMD}^2(\mu, \nu; k)$ for distributions $\mu, \nu$ with support $S_\mu, S_\nu$ if for some $\alpha \in (0, 2)$

$$\sup_{\boldsymbol{x} \in S_\mu, \boldsymbol{y} \in S_\nu} \|\boldsymbol{x} - \boldsymbol{y}\|_2 \leq \frac{\log(\gamma^{2-\alpha})}{(\gamma^2 - 1)h^2}.$$

The implication of the corollary is that when the approximation of the value distribution has bounded support, RBF's $h$-parameter can be chosen such that convergence is guaranteed. Conversely, if the support is unbounded, there is no guarantee of convergence. With finite rewards and $\gamma \in [0, 1)$ the true value distribution will always have a bounded support. This means that a model with bounded support does not necessarily limit the model's ability to approximate the true value distribution. Choosing $h$ small enough to satisfy this condition, however, makes the kernel very flat near 0, hence reducing the kernel's power to differentiate between samples close together.

We noticed that for all the Atari games that MMDQN (Nguyen-Tang et al., 2021) released videos of the $Z$-distribution was uni-modal. Multi-modal modeling of the $Z$-distribution is important when making risk-adjusted actions, where it may be worth choosing an action with a slightly lower expected return if it means not risking a really low return. Although our experiments do not take risk into account when performing actions, we hypothesize a uni-modal approximation could still be a limiting factor for performance in the Atari benchmarks. Therefore, we designed a procedure to evaluate MMD's ability to fit multi-modal distributions with the RBF kernel. The procedure is described in Algorithm 1.

---

**Algorithm 1** Procedure for evaluating the ability of MMD with kernel $k$ to approximate a distribution.

---

**Require:** $k, P, \{x\}_{1:N}, T$
   AD-statistics $\leftarrow Array[1 : T]$
   **for** $t = 1 : T$ **do**
      $y_i \sim P$       for $i = 1, \ldots, N$
      $\{\Delta x\}_{1:N} \leftarrow \nabla_x \widehat{\text{MMD}}_b \left(\{x\}_{1:N}, \{y\}_{1:N}; k\right)$
      $\{x\}_{1:N} \leftarrow Optimiser(\{x\}_{1:N}, \{\Delta x\}_{1:N})$
      AD-statistics$[t] \leftarrow$ Anderson-Darling$(\{x\}_{1:N}, P)$
   **end for**
   **return** AD-statistics

---

The procedure starts with some pseudo-samples $\{x\}_{1:N}$ and a target distribution $P$. Over $T$ gradient updates, the distribution represented by the pseudo samples should move closer to $P$. The Anderson-Darling test (Anderson & Darling, 1952) tests whether a given sample comes from a distribution. We use this test to evaluate the effectiveness of the kernels to fit the distribution. The test scores samples on a scale of 0 to 1, where a low score indicates that it is unlikely that the samples come from the target distribution.

For the Anderson-Darling test experiment, we chose a Gaussian mixture model with two components, $\mathcal{N}(-2, 0.2)$ and $\mathcal{N}(2, 0.2)$ with weights 0.2 and 0.8 respectively. For comparison, we tested the multiquadric kernel (MQ) with $h = 1$, $h = 10$, and $h = 100$. For the RBF we used the same kernel that Nguyen-Tang et al. (2021) used in their Atari experiments, viz. Equation 14. Search for a more effective RBF yielded no positive results. Even though such a kernel may exist, our results show that it is not trivial to find such a kernel, and suggest that performance can suffer as a result. The results of the experiment is shown in Figure 2. A learning rate of 0.1 was used for MQ and a learning rate of 0.01 was used for RBF. MQ was scaled by $h^{-1}$ to avoid the magnitude of the gradients growing significantly with $h$.

## 5 Experiments

To evaluate the performance of the multiquadric kernel, we conducted experiments on 8 Atari games from the Arcade Learning Environment (ALE). Our goal was to see whether our algorithm was able to achieve results similar to those obtained by MMDQN with the RBF kernel, a state-of-the-art distributional reinforcement learning algorithm, while also demonstrating the benefits of using a kernel with theoretical guarantees.

To show that we are not handpicking games that the multiquadric kernel performs better on, we use the 6 Atari games Nguyen-Tang et al. (2021) used for tuning (Breakout, Assault, Asterix, MsPacman, Qbert, and BeamRider). We have selected two additional games, one where MMDQN performed significantly better than QR-DQN (Tutankham), and one where it performed significantly worse (SpaceInvaders). We are interested in whether the multiquadric kernel, with its contraction property, will outperform the RBF kernel in environments where the RBF kernel struggles. By inspecting the original results presented in the MMDQN paper Nguyen-Tang et al. (2021), we see that most of the

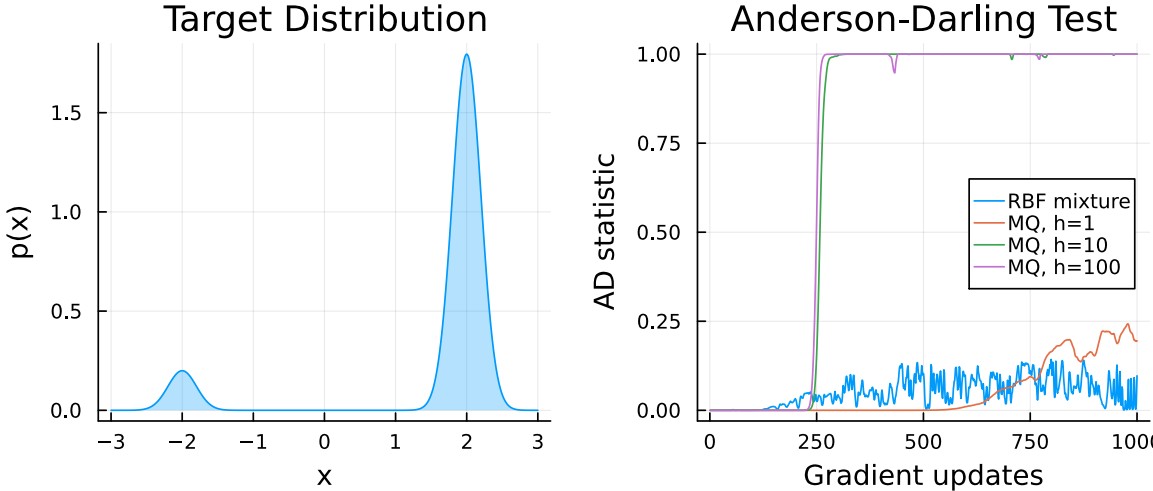

Figure 2: Figure showing target distribution and p-value for the Anderson-Darling test with the null hypothesis that the samples are drawn from $P$ against the alternative hypothesis that the samples are not drawn from $P$.

environments in which MMDQN significantly outperforms QR-DQN are part of the 6 games used for tuning. This suggests that appropriate kernel parameters are important for performance.

To ensure that we do not involuntarily skew the experiments in favor of the multiquadric kernel, we used the original implementation of MMDQN by Nguyen-Tang et al. (2021), and all except the kernel parameters remained the same. Equation 14 defines the parameters used for the RBF mixture kernel. We investigate two parameter values for the multiquadric kernel, $h = 1$ and $h = 10$. The results in Figure 3 show similar performance between the multiquadric and RBF kernel, despite the hyperparameter optimization that has been done for the RBF kernel parameters over these games. This could explain the slightly worse performance in Asterix and Assault. Interestingly, the RBF kernel performed significantly worse on SpaceInvaders. Although no definitive conclusions can be made from this, it does support our argument that the performance is more sensitive to RBF kernel parameters. Raw result data for MQ is available at **https://github.com/ludvigk/MQ-MMDRL**. The numbers for RBF and QR-DQN are taken from Nguyen-Tang et al. (2021), and are available along with the implementation of MMDQN at **https://github.com/thanhnguyentang/mmdrl**.

For more evidence that MQ performs on par with RBF, we have presented human-normalized agregates in Figure 5. As suggested by Agarwal et al. (2021), we have used the more robust interquartile mean (IQM) as an aggregate, rather than a the mean or median. Although these results suggest that MQ performs significantly better, because we are only evaluating the methods on 8 environments, the results for RBF are heavily punished by its poor performance on SpaceInvaders, despite the use of the more robust IQM statistic. In combination with the results presented in Figure 3, we believe that these results show that MQ performs at least on par with RBF.

To evaluate the robustness of the MQ kernel compared to the RBF kernel for its application in MMDQN, we focused on Qbert, one of the 8 Atari games from Figure 3, where a mixture of RBF and MQ showed approximately equal performance. Figure 4 presents the highest human-normalized scores achieved with each kernel. The results unambiguously demonstrate that non-mixture RBF lags behind MQ, providing solid support for the assertion that RBF's performance is significantly more sensitive to hyperparameter tuning.

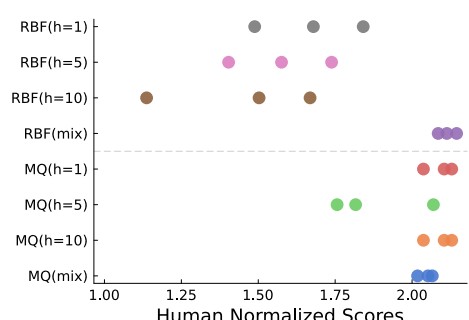

Figure 4: Comparison of highest human-normalized scores for MMDQN on Qbert using different kernels (3 Seeds).

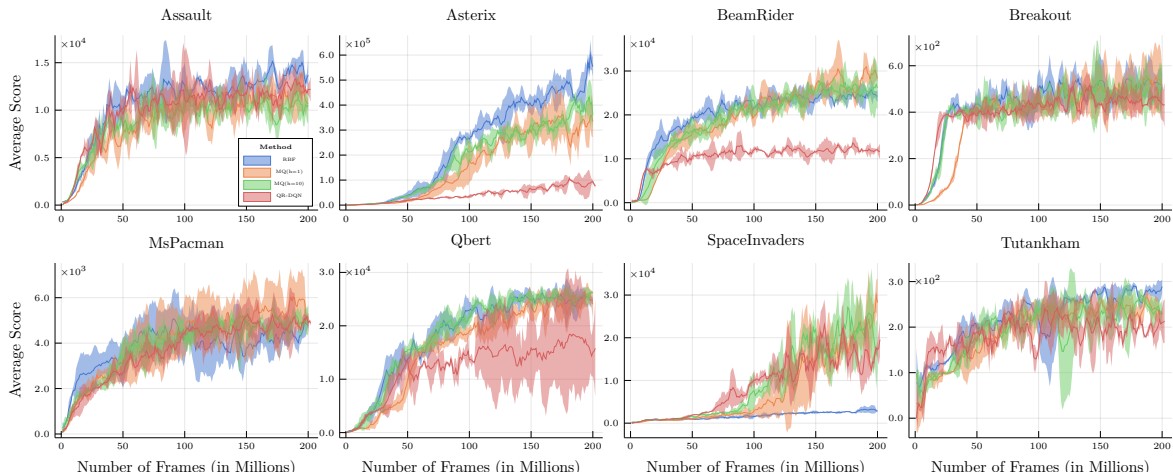

Figure 3: Training curves for QR-DQN and MMDQN with the MQ and RBF on 8 Atari 2600 games. Curves for MMDQN are averaged over 3 seeds and smoothed over a sliding window of 5 iterations. QR-DQN is averaged over 2 seeds. 95% confidence intervals are shown. Reference values for QR-DQN and RBF are from Nguyen-Tang et al. (2021).

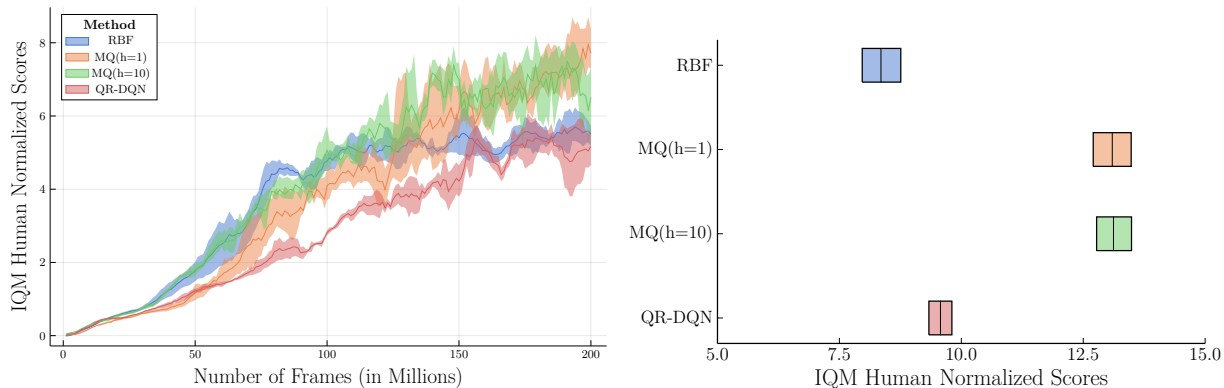

Figure 5: **Left** IQM human-normalized scores as a function of number frames measured. To make the graph more readable, a running mean of size 10 for each seed was used. Shaded regions show pointwise 95% percentile stratified bootstrap CIs. **Right** IQM human-normalized scores with 95% percentile stratified bootstrap CIs of best evaluation for each seed. Reference values for QR-DQN and RBF are from Nguyen-Tang et al. (2021).

## 6 Discussion and Conclusions

We have explored the use of the multiquadric kernel for distributional reinforcement learning in this paper. We have shown that the multiquadric kernel has several key theoretical properties that have been studied in this context:

1. The distributional Bellman operator is a contraction in MMD with the multiquadric kernel.

2. MMD is a metric under the multiquadric kernel, which means that the MMD between two random variables can only be zero when they are equal in distribution.

There are several reasons why a theoretically sound kernel might be preferred over a kernel that performs similarly but lacks theoretical guarantees:

- The contraction property provides a rigorous foundation for the algorithm and can give us confidence in the performance and behavior of the algorithm under different conditions. This is especially important in reinforcement learning, where the agent is learning to interact with and make decisions in a complex and potentially unknown environment.

- The properties discussed in section 3 can help to ensure that the algorithm is well behaved and robust and can help to avoid pathological or degenerate cases that might arise with kernels that lack the metric or contraction property.

- The discussed properties, particularly the metric and contraction properties, can facilitate the development of new techniques and variations on the algorithm by providing a clear set of assumptions and guarantees that can be built upon or modified.

- MMD with MQ kernel's improved ability to model bimodal distribution can be critical for risk-sensitive policies where low returns are particularly detrimental, such as in investment management.

Overall, while it is certainly important to consider the practical performance of a kernel, a theoretically sound kernel can offer a number of additional benefits that can make it a compelling choice for moment-matching distributional reinforcement learning. Additionally, our experimental results show that the multiquadric kernel is stable with regard to parameter changes and that a mixture kernel is not necessary to achieve good results.

Furthermore, through Corollary 4.1, we have provided insight into the practical performance of the RBF kernel. The relationship between convergence, the parameter $h$, and the support of value distributions can provide insight into tuning $h$ for each task. Alternatively, it could provide the basis for an algorithm that learns and changes $h$ during training. Such an algorithm was proposed as future work by Nguyen-Tang et al. (2021). It is possible that the RBF kernel may encounter difficulties in such a scenario due to the absence of a convergence guarantee and that the multiquadric kernel may be a more suitable alternative.

Finally, our results demonstrate the potential of the multiquadric kernel for distributional reinforcement learning. Its contraction and metric property make it a promising choice for a wide range of reinforcement learning tasks. Further research is needed to fully explore the capabilities of the multiquadric kernel and to compare its performance to other kernels in more diverse and challenging environments.

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

# A   Proofs

**Lemma A.1.** Let $k_h$ be the multiquadric kernel, then $\widehat{\mathrm{MMD}}_b(\cdot, \cdot; k_h)$ is smooth.

*Proof.* The multiquadric is infinitely smooth, i.e. is infinitely differentiable. We can see this by looking at its derivatives. First, for $n = 1$ we have

$$\frac{d}{dx} - \sqrt{1 + h^2 x^2} = -h^2 x (1 + h^2 x^2)^{-\frac{1}{2}},$$

and for $n > 1$

$$\frac{d^n}{dx^n} - \sqrt{1 + h^2 x^2} = h^n P_{n-2}(x)(1 + h^2 x^2)^{-n + \frac{1}{2}},$$

where $P_n$ is a polynomial of degree $n$. Since $\mathrm{MMD}_b^2(\cdot, \cdot; k_h)$ is a finite sum of infinitely smooth functions, it is also infinitely smooth.

According to Alekseevsky et al. (1998), non-negative infinitely smooth functions always have a smooth square root. $\mathrm{MMD}_b^2(\cdot, \cdot; k_h)$ is non-negative (Gretton et al., 2012) and as shown above, infinitely smooth, hence $\mathrm{MMD}_b(\cdot, \cdot; k_h)$ must be smooth.  □

**Lemma A.2.** Let $k_h$ be the MQ, and $\widehat{\mathrm{MMD}}_b^2(\{\boldsymbol{x}_i\}_n, \{\boldsymbol{y}_i\}_n; k_h) : A \times B \to \mathbb{R}$, where $A$ and $B$ are sets of real values $\boldsymbol{x}_{1:n} \in \mathbb{R}^d$ and $\boldsymbol{y}_{1:n} \in \mathbb{R}^d$ respectively. Then $\widehat{\mathrm{MMD}}_b^2$ is Lipschitz continuous wrt $A$ and $B$.

*Proof.* The first derivative of the MQ kernel is bounded,

$$\left\| \frac{\partial}{\partial \boldsymbol{x}} - \sqrt{1 + h^2 \|\boldsymbol{x} - \boldsymbol{y}\|_2^2} \right\|_2 = \left\| -h^2 (\boldsymbol{x} - \boldsymbol{y}) \left(1 + h^2 (\boldsymbol{x} - \boldsymbol{y})^2\right)^{-\frac{1}{2}} \right\|_2 \leq \|-h\mathbf{1}_d\|_2 = \sqrt{d} h.$$

Because $\widehat{\mathrm{MMD}}_b^2(\{\boldsymbol{x}_i\}, \{\boldsymbol{y}_i\}; k_h)$ is a finite sum of functions with bounded gradient, its gradient is necessarily also bounded.  □

**Lemma A.3.** $f(t; a, b) := \sqrt{a + t^2} - \sqrt{b + t^2}$ is positive definite for $a > b$.

*Proof.* Let $g(t) := f(\sqrt{t}) = \sqrt{a + t} - \sqrt{b + t}$. We use that a function $f(t)$ is positive definite if and only if $g(t)$ is completely monotonic. By induction, it follows that

$$\frac{d^n}{dt^n} \sqrt{c + t} = (c + t)^{1/2 - n} \left(\frac{3}{2} - n\right)_n,$$

where $(\cdot)_n$ is the Pochhammer symbol. Hence,

$$\frac{d^n}{dt^n} \sqrt{a + t} - \sqrt{b + t}$$
$$= (a + t)^{1/2 - n} \left(\frac{3}{2} - n\right)_n - (b + t)^{1/2 - n} \left(\frac{3}{2} - n\right)_n$$
$$= \left((a + t)^{1/2 - n} - (b + t)^{1/2 - n}\right) \left(\frac{3}{2} - n\right)_n.$$

The first term is positive for $n = 0$ and negative otherwise. The second term is positive for $n = 0$ and has sign $(-1)^{n+1}$ otherwise. It is therefore evident that $g(t)$ is completely monotonic and thus $f(t)$ is positive definite (Lemma 3.2).  □

**Definition A.1.** $(f)_{\#}\mu$ is the distribution of a random variable $f(X)$, with $X \sim \mu$. This is known as the pushforward measure of $\mu$ by $f$.

**Theorem A.1.** The distributional Bellman operator is a contraction in MMD with the multiquadric kernel.

*Proof.* Let $k$ be the multiquadric kernel, i.e.

$$k(\boldsymbol{x}, \boldsymbol{y}) = \psi(\boldsymbol{x} - \boldsymbol{y}) = -\sqrt{1 + h^2 \|\boldsymbol{x} - \boldsymbol{y}\|_2^2}.$$

Nguyen-Tang et al. (2021) prove that for the distributional Bellman operator to be a contraction it is sufficient to show that for all $\gamma \in (0, 1)$

$$\mathrm{MMD}^2\left((f_{r,\gamma})_{\#}\mu, (f_{r,\gamma})_{\#}\nu; k\right) \leq \gamma \, \mathrm{MMD}^2(\mu, \nu; k),$$

where $(f_{r,\gamma})_{\#}\mu$ denotes the pushforward measure of $\mu$ by $(f_{r,\gamma})_{\#}$. Nguyen-Tang et al. (2021, Lemma 4) states that this holds with equality for all scale sensitive and shift-invariant kernels $k$, i.e. whenever $k(r+\gamma x, r+\gamma y) = \gamma^\alpha k(x, y)$ for some $\alpha > 0$. We will show that this condition holds for the multiquadric kernel.

First, we will define

$$\tilde{\psi}_\gamma(\boldsymbol{z}) := -\sqrt{\frac{1}{\gamma^2} + h^2 \|\boldsymbol{z}\|_2^2},$$

with

$$\tilde{k}_\gamma(\boldsymbol{x}, \boldsymbol{y}) := \tilde{\psi}_\gamma(\boldsymbol{x} - \boldsymbol{y}).$$

Note that $\psi(\gamma \boldsymbol{z}) = \gamma \tilde{\psi}_\gamma(\boldsymbol{z})$.

$$
\begin{aligned}
\mathrm{MMD}^2\left((f_{r,\gamma})_{\#}\mu, (f_{r,\gamma})_{\#}\nu; k\right) &= \int\int k(\boldsymbol{z}, \boldsymbol{z}')(f_{r,\gamma})_{\#}\mu(d\boldsymbol{z})(f_{r,\gamma})_{\#}\mu(d\boldsymbol{z}') \\
&\quad + \int\int k(\boldsymbol{w}, \boldsymbol{w}')(f_{r,\gamma})_{\#}\nu(d\boldsymbol{w})(f_{r,\gamma})_{\#}\nu(d\boldsymbol{w}') \\
&\quad - 2\int\int k(\boldsymbol{w}, \boldsymbol{w}')(f_{r,\gamma})_{\#}\mu(d\boldsymbol{z})(f_{r,\gamma})_{\#}\nu(d\boldsymbol{w}) \\
&= \int\int k(r+\gamma\boldsymbol{z}, r+\gamma\boldsymbol{z}')\mu(d\boldsymbol{z})\mu(d\boldsymbol{z}') + \int\int k(r+\gamma\boldsymbol{w}, r+\gamma\boldsymbol{w}')\nu(d\boldsymbol{w})\nu(d\boldsymbol{w}') \\
&\quad - 2\int\int k(r+\gamma\boldsymbol{z}, r+\gamma\boldsymbol{w})\mu(d\boldsymbol{z})\nu(d\boldsymbol{w}) \\
&= \int\int \psi(\gamma\boldsymbol{z} - \gamma\boldsymbol{z}')\mu(d\boldsymbol{z})\mu(d\boldsymbol{z}') + \int\int \psi(\gamma\boldsymbol{w} - \gamma\boldsymbol{w}')\nu(d\boldsymbol{w})\nu(d\boldsymbol{w}') \\
&\quad - 2\int\int \psi(\gamma\boldsymbol{z} - \gamma\boldsymbol{w})\mu(d\boldsymbol{z})\nu(d\boldsymbol{w}) \\
&= \int\int \gamma\tilde{\psi}_\gamma(\boldsymbol{z} - \boldsymbol{z}')\mu(d\boldsymbol{z})\mu(d\boldsymbol{z}') + \int\int \gamma\tilde{\psi}_\gamma(\boldsymbol{w} - \boldsymbol{w}')\nu(d\boldsymbol{w})\nu(d\boldsymbol{w}') \\
&\quad - 2\int\int \gamma\tilde{\psi}_\gamma(\boldsymbol{z} - \boldsymbol{w})\mu(d\boldsymbol{z})\nu(d\boldsymbol{w}) \\
&= \gamma \, \mathrm{MMD}^2(\mu, \nu; \tilde{k}_\gamma)
\end{aligned}
$$

Showing that $\mathrm{MMD}^2(\mu, \nu; \tilde{k}_\gamma) \leq \mathrm{MMD}^2(\mu, \nu; k)$, or equivalently $\mathrm{MMD}^2(\mu, \nu; k) - \mathrm{MMD}^2(\mu, \nu; \tilde{k}_\gamma) \geq 0$ is therefore sufficient.

$$\text{MMD}^2(\mu, \nu; k) - \text{MMD}^2(\mu, \nu; \tilde{k}_\gamma) = \left( \int \int \psi(\boldsymbol{z} - \boldsymbol{z}') \mu(d\boldsymbol{z}) \mu(d\boldsymbol{z}') + \int \int \psi(\boldsymbol{w} - \boldsymbol{w}') \nu(d\boldsymbol{w}) \nu(d\boldsymbol{w}') \right.$$

$$- 2 \int \int \psi(\boldsymbol{z} - \boldsymbol{w}) \mu(d\boldsymbol{z}) \nu(d\boldsymbol{w}) \right) - \left( \int \int \tilde{\psi}_\gamma(\boldsymbol{z} - \boldsymbol{z}') \mu(d\boldsymbol{z}) \mu(d\boldsymbol{z}') \right.$$

$$\left. + \int \int \tilde{\psi}_\gamma(\boldsymbol{w} - \boldsymbol{w}') \nu(d\boldsymbol{w}) \nu(d\boldsymbol{w}') - 2 \int \int \tilde{\psi}_\gamma(\boldsymbol{z} - \boldsymbol{w}) \mu(d\boldsymbol{z}) \nu(d\boldsymbol{w}) \right)$$

$$= \int \int \psi(\boldsymbol{z} - \boldsymbol{z}') - \tilde{\psi}_\gamma(\boldsymbol{z} - \boldsymbol{z}') \mu(d\boldsymbol{z}) \mu(d\boldsymbol{z}')$$

$$+ \int \int \psi(\boldsymbol{w} - \boldsymbol{w}') - \tilde{\psi}_\gamma(\boldsymbol{w} - \boldsymbol{w}') \nu(d\boldsymbol{w}) \nu(d\boldsymbol{w}')$$

$$- 2 \int \int \psi(\boldsymbol{z} - \boldsymbol{w}) - \tilde{\psi}_\gamma(\boldsymbol{z} - \boldsymbol{w}) \mu(d\boldsymbol{z}) \nu(d\boldsymbol{w})$$

$$= \text{MMD}^2(\mu, \nu; k - \tilde{k}_\gamma)$$

Since $\text{MMD}^2$ is non-negative for conditionally positive definite kernels, it is sufficient to show that

$$\psi(\boldsymbol{z}) - \tilde{\psi}_\gamma(\boldsymbol{z})$$

is conditionally positive definite for $\gamma \in (0, 1)$. Lemma 3.4 shows that

$$\psi(\boldsymbol{z}) - \tilde{\psi}_\gamma(\boldsymbol{z}) = \sqrt{\frac{1}{\gamma^2} + h^2 \|\boldsymbol{z}\|_2^2} - \sqrt{1 + h^2 \|\boldsymbol{z}\|_2^2}$$

is positive definite for $\gamma \in (0, 1)$, and therefore conditionally positive definite, which concludes the proof. $\qquad \square$

**Corollary A.1.** Let $k(\boldsymbol{x}, \boldsymbol{y}) = \exp(-h^2 \|\boldsymbol{x} - \boldsymbol{y}\|_2^2)$. Then the distributional Bellman operator is a contraction in $\text{MMD}^2(\mu, \nu; k)$ for distributions $\mu, \nu$ with support $S_\mu, S_\nu$ such that for some $0 < \alpha < 2$

$$\sup_{\boldsymbol{x} \in S_\mu, \boldsymbol{y} \in S_\nu} \|\boldsymbol{x} - \boldsymbol{y}\|_2 \leq \frac{\log(\gamma^{2-\alpha})}{(\gamma^2 - 1)h^2}. \tag{17}$$

*Proof.* Utilizing the proof for Theorem 3.1, we note that all we have to do is show that

$$\gamma^\alpha k(\boldsymbol{x}, \boldsymbol{y}) - k(\gamma \boldsymbol{x}, \gamma \boldsymbol{y})$$

is conditionally positive definite when Equation (17) holds.

By Lemma 3.2, this means that

$$-\frac{d}{dz} \gamma^\alpha \exp(-h^2 z) - \exp(-h^2 \gamma^2 z) = \gamma^\alpha h^2 \exp(-h^2 z) - h^2 \gamma^2 \exp(-h^2 \gamma^2 z)$$

is conditionally positive definite. Let us begin by showing that the first derivative is negative.

$$h^2 \gamma^\alpha h^2 \exp(-h^2 z) - \gamma^2 \exp(-h^2 \gamma^2 z) \geq 0$$

$$\iff \exp(-h^2 z) - \gamma^{2-\alpha} \exp(-h^2 \gamma^2 z) \geq 0$$

$$\iff -h^2 z \geq \log(\gamma^{2-\alpha}) - h^2 \gamma^2 z$$

$$\iff h^2 \gamma^2 z - h^2 z \geq \log(\gamma^{2-\alpha})$$

$$\iff h^2 (\gamma^2 - 1) z \geq \log(\gamma^{2-\alpha})$$

$$\iff z \leq \frac{\log(\gamma^{2-\alpha})}{(\gamma^2 - 1)h^2} \qquad \text{(This is the requirement in Equation 17)}$$

Now we show that for $n \geq 1$

$$(-1)^n \frac{d}{dz^n} \gamma^\alpha \exp(-h^2 z) - \exp(-h^2 \gamma^2 z) \geq 0$$

whenever

$$-\frac{d}{dz} \gamma^\alpha \exp(-h^2 z) - \exp(-h^2 \gamma^2 z) \geq 0.$$

For $\gamma \in (0, 1)$ we have

$$
\begin{aligned}
(-1)^n \frac{d}{dz^n} \gamma^\alpha \exp(-h^2 z) - \exp(-h^2 \gamma^2 z) &= \gamma^\alpha h^{2n} \exp(-h^2 z) - h^{2n} \gamma^{2n} \exp(-h^2 \gamma^2 z) \\
&= h^{2n-1} \left( \gamma^\alpha h^2 \exp\left(-h^2 z\right) - h^2 \gamma^{2n} \exp\left(-h^2 \gamma^2 z\right) \right) \\
&\geq h^{2n-1} \left( \gamma^\alpha h^2 \exp\left(-h^2 z\right) - h^2 \gamma^2 \exp\left(-h^2 \gamma^2 z\right) \right).
\end{aligned}
$$

Given that $\gamma^\alpha h^2 \exp\left(-h^2 z\right) - h^2 \gamma^2 \exp\left(-h^2 \gamma^2 z\right) \geq 0$ this shows that the $(-1)^n \frac{d^n}{dz^n} \gamma^\alpha \exp(-h^2 z) - \exp(-h^2 \gamma^2 z)$ is also greater than or equal to zero. $\qquad\square$

