# OpenReview forum: "The Multiquadric Kernel for Moment-Matching Distributional Reinforcement Learning"
_TMLR — Accepted by TMLR_

### Review · Reviewer_376H · 2023-06-14

**Summary Of Contributions:**

The authors build upon Nguyen-Tang et al. (2021)’s work on distributional RL using the MMD. They combine Nguyen-Tang et al. (2021)’s algorithm MMDQN with a new kernel, the multiquadratic kernel, and demonstrate that it expresses various theoretical guarantees. They evaluate it empirically against MMDQN with the RBF kernel used in (Nguyen-Tang et al., 2021), and obtain reasonably strong results. They complement this with theoretical support for the multiquadratic kernel, by demonstrating that the MMD with this kernel is smooth, has bounded gradients, and the distributional Bellman operator is a contraction under the MMD with this kernel.

**Audience:**

Yes

**Broader Impact Concerns:**

The paper does not contain a "Broader Impact Statement". Despite this the contribution is mainly fundamental and I think this is OK.



**Claims And Evidence:**

No

**Requested Changes:**

I would like to see my weakness listed above to be addressed, as I consider them of high importance.

**Strengths And Weaknesses:**

**Strengths**:
- The experiments demonstrate strong empirical results, and are relatively statistically significant (8 Atari games x 3 seeds each).
- They make a novel observation that using the MMD with an RBF kernel was not able to accurately fit multimodal distributions.


**Weaknesses**:
- I have serious doubts regarding Corollary 4.1. The inequality seems impossible to hold: the LHS is always positive as it is a norm, while the RHS is negative (the numerator is positive, $h^2$ is positive, and $\gamma^2-1$ is negative since $\gamma \in (0,1)$, so the whole term is negative). As the corollary states that the contraction property holds whenever we have that LHS $\leq$ RHS, this statement appears to trivially never hold.

- The fact that properties (1)-(4) (using the naming of the paper) holds for the multiquadratic kernel is one of the main contributions of the paper. The authors did not prove properties (1) nor (2) however, and claim “Properties 1 and 2 can be verified by inspection”. These properties do not appear trivial to me however, and in fact I am doubtful they both hold. To this end I request the authors to include a proof of these properties in the paper (in the appendix is fine).

- Definition 4.2 as presented is not the definition of positive-definite kernel used in the literature (e.g. Gretton). It seems the authors define a positive definite kernel to be the negative of a distance of negative type as introduced in (https://arxiv.org/pdf/1106.5758.pdf). Of course the authors can define things as they wish, but I believe that by using the commonplace term “positive definite kernel” with a different definition misleads the reader and should be changed. Note that the multiquadratic kernel k_h they introduce is not a proper positive definite kernel in the traditional sense (the one used in gretton…) - a simple counterexample is that k(1,1) < 0, while this must be positive for any p.d. Kernel.

**Minor comments/typos**:

- In Definition 4.2, the terms are sums of $cc^*$ – I assume this should be $c^* c$ as $c c^*$ is a matrix.
- After introducing the optimal Bellman operator and demonstrating that $(T^*)^kQ\to Q^*$ as $k\to\infty$ for any $Q$, the authors state "This means that if we adopt π in a “GLIE manner”, then $Q^π = Q^*$ in the limit". I don't believe this discussion is relevant here, as the convergence of applications of the Bellman operator does not have anything to do with exploration as presented, and this discussion appears out of place to me.
- There are a number of places in the text which would benefit from a careful re-reading, there are a number of minor typos and grammar mistakes.
- You refer to the multiquadratic kernel in Section 3 before defining it in Section 4.
- The equations MMD_b and MMD_u referenced were both introduced by [Gretton et al. (2012)](https://www.jmlr.org/papers/volume13/gretton12a/gretton12a.pdf), I would recommend citing.

---

> ### Author Response · Authors · 2023-06-16
> **Response to Reviewer 376H**
>
> Thank you for your detailed review. We will try to address your main concerns first:
>
> ---
>
> [RE 1]
>
> First, you mention you have doubts about Corollary 4.1 because the LHS is always positive and the RHS is negative. For the RHS both the numerator and denominator are negative. We will rephrase to say "for some $0 < \alpha < 2$" to make it clear that $\log(\gamma^{2-\alpha})$ is always negative.
>
> ---
>
> [RE 2]
>
> We acknowledge that the statement “Properties 1 and 2 can be verified by inspection” is too brief. What we meant to say is that the properties can be verified by inspection for $MMD_b^2$. The statements are still true for $MMD_b$, but these are not trivial proofs. We will include proof of these properties in the revised version of the paper, addressing any doubts or concerns regarding their validity. Additionally, the statements
>
> > (1) $\mathrm{MMD}(\cdot,\cdot,k_h)$ is smooth
> >
> > (2) $\sup_{x \in \mathbb{R}^n}\rVert \frac{\partial}{\partial x} \mathrm{MMD}(x, y; k_h) \lVert \leq C_h$
>
> will be changed to
>
> > (1) $\mathrm{MMD_b}(x,\cdot,k_h)$ is smooth with respect to $x$.
> >
> > (2) $\sup_{x \in \mathbb{R}^n}\rVert \frac{\partial}{\partial x} \mathrm{MMD_b}(x, y; k_h) \lVert < C_h < \infty$.
>
>
> Proof of property 1:
>
> > The multiquadric kernel, $ -\sqrt{1 + \lVert x - y\rVert^2}$, is infinitely smooth, which means that $MMD_b^2$, being a finite sum of infinitely smooth functions, is infinitely smooth. According to Dmitri Alekseevsky (1998) _Choosing roots of polynomials smoothly_, infinitely smooth functions always have a smooth square root, hence $\mathrm{MMD_b}$ is smooth.
>
>
> Proof of property 2: (This proof will be cleaned up before it is added to the appendix)
> > The 2nd derivative of the multiquadric kernel, $\left\lVert \frac{\partial}{\partial x}  -\sqrt{1 + \lVert x - y\rVert^2} \right\rVert =  \left\lVert - \frac{h^2}{h^2(x-y)^2\sqrt{1 + h^2 (x-y)^2}+\sqrt{1 + h^2(x-y)^2}} \right\rVert \leq h^2$.
> > This means that the absolute value of the second derivative of $\mathrm{MMD_b^2}$ is bounded by $4h^2$, since it is the sum of 4 terms bounded by $h^2$.
> > This implies that $\left\lvert \frac{\frac{\partial}{\partial x} \mathrm{MMD_b}^2(x, y; k_h)}{2\mathrm{MMD_b}(x, y; k_h)} \right\rvert \leq 2h$, from which Lipschitz continuity of $\mathrm{MMD_b}$ follows.
> > Suppose by contradiction there is a point where the inequality does not hold. Then $\frac{\partial}{\partial x} \mathrm{MMD_b}^2(a, b; k_h) > 4h^2 \mathrm{MMD_b}(a, b; k_h)$. From the bound $\mathrm{MMD_b^2} \leq 4h^2$ we know that $\mathrm{MMD_b}^2(x, b; k_h) \leq 4h^2 x^2 + \frac{\partial}{\partial x}\mathrm{MMD_b}^2(x, b; k_h)\big\vert_a + \mathrm{MMD_b}^2(a, b; k_h)$. The discriminant of the polynomial is $\mathrm{MMD_b}^2(a, b; k_h) ^2-16h^2 \mathrm{MMD_b}^2(a, b; k_h) > 0$, which implies that the polynomial has roots and thus $\mathrm{MMD_b}^2$ becomes negative, which is a contraction.
>
> ---
>
> [RE 3]
>
> With respect to Definition 4.2, it is indeed not the definition of a positive definite kernel, but of **conditionally** positive definite (cpd) kernels. This term was used e.g. by Schölkopf and Smola (2002): «Learning with Kernels». The class of cpd kernels extends the positive definite kernels, and while the multi-quadric kernel is not positive definite, we only need it to be in the cpd kernels to prove that the Bellman operator with MMD using the multi-quadric kernel is a contraction.
>
> ---
>
> I would also like to thank you for the minor comments/typos. We agree with all of them and will update the article accordingly.

---

### Review · Reviewer_KzRQ · 2023-06-15

**Summary Of Contributions:**

The authors analyse the properties of multiquadratic kernel for moment matching distributional RL. They show it holds 4 desired properties:
smoothness, bounded gradient, metric properties and contraction. They also show that empirically it works just as fine as the standard alternative.

**Audience:**

Yes

**Claims And Evidence:**

Yes

**Requested Changes:**

Small clarity issues (none would affect the recommendation for acceptance) but I think would help the paper:

1. “Our contribution is mainly of a theoretical nature, presenting the first formally sound kernel for moment-matching distributional reinforcement learning with good practical performance. “

This sentence is two sentences in one and thus confusing: 1. The contribution is the first theoretical result 2. good practical performance (this already exists in the previous sentence).

2. You relate to MQ before it is defined in the end of Section 3 and in Figure 1.

3. Introduction and related works very long making the paper survey-like.

4. In Section 4 the order stuff are written in is confusing. You assume MQ is a contraction, but then you prove it, same for metric properties. The table is after the main points. The easiest fix for me would be to reference the table and theorems as soon as you talk about things related to them.

5. In concisely formulated, Ch is not defined.

6. In Definition 4.1, psi is not defined.

7. Figure 3 – missing axis names (y=episodic reward, x=steps ?)


**Strengths And Weaknesses:**

+ A novel result that has not been addressed yet.

+ Half survey on the topic of kernels for MMDs.

+ I liked that the authors showed the practical results were not compromised by their proposed metric.

- I'm not sure how many researchers will take interest in the result, that are very limited in scope.

- The significance and novelty, while exist, are very modest - previous kernels comply to almost all 4 properties, and as the authors suggested themselves smoothing the l1 near 0 is very natural thing to do that was already done in multiple other cases.

- Some clarity issues (see requested changes). While the writing was overall clear, it wasn't organized enough.

---

> ### Author Response · Authors · 2023-06-23
> **Response to Reviewer KzRQ**
>
> Dear Reviewer,
>
> Thank you for taking the time to review our paper. We appreciate your valuable feedback and suggestions for improving the clarity and organization of our work. We agree with the requested changes. We have addressed each of your concerns in our revised manuscript.
>
> Firstly, we acknowledge the clarity issue in the sentence: "Our contribution is mainly of a theoretical nature, presenting the first formally sound kernel for moment-matching distributional reinforcement learning with good practical performance." We have separated it into two distinct sentences to eliminate confusion and clearly state the theoretical nature of our contribution and the practical performance of the proposed kernel.
>
> In Section 4, we have reorganized the content to improve the flow and address your concerns. We now refer to the table and theorems as soon as they become relevant.
>
> Furthermore, we have defined $C_h$ in the concise formulation section and made changes in Definition 4.1 to eliminate a $\psi$, as that was meant to be $f$.
>
> We would like to emphasize the importance of our work. While previous kernels may comply with some of the desired properties, our proposed multiquadratic kernel addresses the challenge of finding appropriate kernels for different environments. This makes our method highly viable for a wide range of applications in this domain. Furthermore, we see the potential for our method to be combined with other advancements in the field. By addressing the issue of appropriate kernel selection, we pave the way for further application of moment-matching distributional RL.
>
> Once again, we appreciate your constructive feedback and suggestions. We have taken them into account during the revision process, and we are confident that the revised manuscript addresses the concerns you raised. We hope you find the changes satisfactory.
> Thank you for your time and consideration.

---

### Review · Reviewer_KxkT · 2023-06-22

**Summary Of Contributions:**

This paper focuses on the moment matching approach to minimizing the distributional Bellman error. The aim of the paper is to propose a theoretically sound measure of difference between two distributions that can be used for this approach. To do so it proposes the multiquadratic kernel, and analyzes its properties when applied to distributional reinforcement learning.

This theoretical analysis shows that the maximum mean discrepancy (MMD) for the multiquadratic kernel is a metric in the space of distributions, and that the distributional Bellman operator is a contraction in MMD with this multiquadratic kernel.

Additionally they also show some conditions under which the distributional Bellman operator is also a contraction in MMD for the RBF kernel.

The experiments modify MMDQN with this multiquadratic kernel and show that it performs just as well or better compared to the RBF kernel.

**Audience:**

Yes

**Claims And Evidence:**

Yes

**Requested Changes:**

The requested changes below are in decreasing order of importance. I am also willing to waive the requirement for any of these based on compelling reasoning against them.

* In Section 3, some analysis of the presented result, a description of the Anderson-Darling test, and what a reader should expect and take away from the section would improve its effectiveness
* In experiments, a comparison with some of the other effective distributional RL techniques such as QR-DQN would give a more holistic view of the proposed kernel beyond just the MMDQN algorithm.
* To showcase the stability of the MQ kernel, perhaps the result of a sweep over `h` for the MQ kernel and for the radius over the RBF kernel could be done to show the relative stability of the MQ kernel?
* Section 3 and Figure 2 discusses the MQ kernel before it is introduced. Rework this section and the next. Perhaps introduce the MQ kernel earlier if the analysis in Section 3 is reliant on comparison with the MQ kernel.
* In Section 4, the paper states: "First, under the multiquadric kernel, maximum mean discrepancy (MMD) is a metric in the family of valid distributions." Clarify that this statement is proved in the paper. As it stands it seems like a statement without any source or proof behind it.
* In paragraph 3, on page 7, the paper states: "On the other hand, if the loss function is smooth, then the gradients will be relatively stable, which allows the model to make more consistent and reliable updates to its parameters during training." Is there a citation to back this claim?
* The list of desirable properties for the MQ kernel on this page (page 7) is also out of order compared to their description in text above it. Reorder them to be consistent, maybe?
* In Definition 4.1, could $\psi$ be described, and the connection of the equation to the function $f$ also be clarified?
* In Definition 4.2, could the significance of the * in $c_j^*$ be clarified?
* Since the results are a comparison over only 3 seeds, perhaps metrics such as the interquartile mean across all the runs or some of the other suggestions put forth in [1] could be used?

References:
[1] Deep Reinforcement Learning at the Edge of the Statistical Precipice https://arxiv.org/abs/2108.13264

**Strengths And Weaknesses:**

## Strengths
* The background section is well written and covers the distributional RL landscape well
* The proposed kernel is an addition to the literature that might be of interest to researchers in the distributional RL community.
* The procedure to evaluate MMD with the RBF kernel to showcase some of its weaknesses crystallizes the requirement for an improved kernel.
* Figure 2 is a good qualitative comparison of the different kernels
* The theoretical analysis of the new kernel gets at important properties desirable of any measure for the distributional Bellman error.
* The analysis to rationalize the empirical success of the RBF kernel is also a worthwhile addition to the manuscript.
* The experiments seem to indicate the the multiquadratic kernel is not worse than the RBF kernel for use in the MMDQN algorithm.

## Weaknesses
* There are certain changes that would make the paper stronger which I list out below
* Apart from the toy example in Section 3 and the results in Figure 2, there is no experiment showcasing the stability of the MQ kernel compared to the RBF kernel. The `SpaceInvaders` result could be construed as such, but improved performance could be attributed to various differences and not just improved convergence.

---

> ### Author Response · Authors · 2023-06-28
> **Response to Reviewer KxkT**
>
> Dear reviewer,
>
> Thank you for your insightful review of our paper.
>
> Regarding the requested changes, we have taken them into careful consideration and made the following revisions:
>
> 1. We will add some explanation of the Andersson-Darling test statistic, what we should expect to see, and how to interpret the results.
> 2. We agree that further comparison in the training curve would be beneficial. We will include training curves for QR-DQN to provide a more comprehensive view and comparison of the proposed kernel.
> 3. A full sweep over h for the MQ kernel could be done, but it is our impression that the current results sufficiently showcase the stability of the kernel.
> If these experiments are a requirement to be accepted for publication, we will of course add them, but this will add several months to the publishing timeline. We will add a sentence stating that Nguyen et al. show that the RBF kernel with various bandwidths performs significantly worse than the mixed kernel.
> 4. We have addressed the issue of concepts and ideas being out of order. The introduction of the MQ kernel has been moved to Section 2.2.4 under the section "kernels." Furthermore, Section 4 has been reworked to introduce properties 1-4 before discussing them in detail.
> 5. We agree that the statement was unclear, and we have revised it according to your suggestion.
> 6. We acknowledge that this statement may be too strong without a citation. We revise the statement to the following, “In many practical scenarios, it is commonly observed that a smoothness property in the loss function tends to lead to gradients that exhibit a greater degree of stability and better performance.” [1,2]
> 7. Thank you for the suggestion to reorder the properties in Section 4. We have implemented this change, and they now follow the same order as they are discussed in the section.
> 8. We have corrected the definition, and $\psi$ is now represented as $f$.
> 9. We have revised the definition to make it clearer and more understandable.
> 10. We appreciate your insight regarding aggregate scores. As we did not conduct experiments across all 57 Atari games, making the scores comparable to those in other papers would be challenging. Therefore, we chose to present the results in a visual plot. Furthermore, the running average and presented standard deviation in the plots are done to make them easily comparable to the plots by Nguyen et al. However, we will include QR-DQN in the plot to provide a quick overview of the performance of MMDQN compared to other DRL methods.
>
> We sincerely appreciate your thorough review and constructive feedback, which have greatly contributed to improving the quality and clarity of our work.
>
> [1] Robust Losses for Learning Value Functions (https://arxiv.org/abs/2205.08464)
> [2] Generalized Huber Loss for Robust Learning and its Efficient Minimization for a Robust Statistics (https://arxiv.org/abs/2108.12627)

---

> > ### Comment · Reviewer_KxkT · 2023-07-01
> > **Revision Request**
> >
> > I thank the authors for addressing the issues I raised in my review.
> > I respond briefly here to some of the points raised in the response:
> > * "A full sweep over $h$ for the MQ kernel" is not necessary. My request was just to show a coarse sweep to show that the MQ kernel is more robust compared to the RBF kernel. This is not a requirement, but I believe it will make the submission stronger
> > * The IQM metric and stratified sampling does not require results on all 57 games. You can run those comparisons just on the games you have tested on. Or is it not possible because you have not run the baseline MMDQN algorithm? In any case, a comparison using the techniques suggested is strongly recommended. I am not convinced as of now that such an evaluation is not possible.
> > * To evaluate whether the rest of the changes made have improved the quality of the paper, I request the authors upload the revision here so that we may evaluate ourselves how many of the requested changes have been incorporated.
> > * Lastly, I would like to look more closely at the review by *Reviewer 376H* and your response to it. A revision addressing their comments will also be helpful in analyzing whether their requests have been addressed.

---

> > > ### Author Response · Authors · 2023-07-06
> > > **Update to Response by Reviewer KxkT**
> > >
> > > * Regarding your first point about a coarse sweep, we suggest choosing one game where both RBF and MQ perform equally well (such as Qbert) and run RBF with $h=1$ and $h=10$ (and perhaps $h=5$ for both MQ and RBF). Would this address your suggestion? To give an idea of the timeframe we are talking about regarding more experiments, we have the capacity to run 7-8 experiments (Atari 200M) per week.
> > >
> > > * Thank you for pointing out that we can still show the aggregate performance as  We have misunderstood what data you suggested be aggregated over. We have now updated the article with the aggregate performance using the IQM metric. With only 8 games, RBF's poor performance on SpaceInvaders is still evident in the aggregate metric, even when using IQM.
> > >
> > > * We have now uploaded a revised version of the paper that we hope adequately addresses all your concerns, as well as those brought forward by _Reviewer 376H_. The revised version does not yet include a sweep over $h$.

---

> > > > ### Comment · Reviewer_KxkT · 2023-07-07
> > > > **Response to Update**
> > > >
> > > > I thank the authors for taking my suggestions into account. The paper seems like a much stronger submission now. My concerns have been addressed.
> > > >
> > > > The authors' suggestion of showcasing the robustness of the MQ kernel with a single experiment but different kernel sizes might be sufficient. But I do recommend running that experiment on the three kernel sizes they have pointed out.

---

> > > > > ### Author Response · Authors · 2023-08-01
> > > > > **Update Regarding Experiment Suggested by Reviewer KxkT**
> > > > >
> > > > > We have uploaded an updated manuscript with the suggested experiment. It provides support for the claim that the MQ kernel is less sensitive to hyperparameter tuning. Thank you for suggesting this experiment.

---

### Decision · Action_Editors · 2023-08-12

**Recommendation:** Accept as is

**Comment:**

The paper considers the MMD approach to distributional reinforcement learning. It proves that Multi-Quadratic kernel has certain desirable properties (being a metric in the space of probabilities, Bellman operator being contraction w.r.t. the kernel, smoothness, and boundedness of derivative). Other commonly used kernels, including RBF, do not satisfy all these properties.

We have two Accepts and one Leaning Accept among reviewers. They find the revisions satisfactory. So I recommend acceptance as is.


**Audience:**

The paper is of interest to a sub-community of researchers (distributional reinforcement learning).

**Claims And Evidence:**

Yes, all reviewers believe that the claims have clear evidence.